



# Development of Low-Cost Air Quality Stations for Next Generation Monitoring Networks: Calibration and Validation of NO2 and O3 Sensors

Alice Cavaliere[1], Lorenzo Brilli[1], Bianca Patrizia Andreini[2,*], Federico Carotenuto[1], Beniamino Gioli[1], Tommaso Giordano[1], Marco Stefanelli[2,*], Carolina Vagnoli[1], Alessandro Zaldei[1], and Giovanni Gualtieri[1]

[1]National Research Council-Institute for BioEconomy (CNR-IBE), Via Caproni 8, 50145 Firenze, Italy
[2]ARPAT, Tuscany Region Environmental Protection Agency, Via Porpora, 22, 50144 Firenze, Italy
*These authors contributed equally to this work.

**Correspondence:** Alice Cavaliere (alice.cavaliere@ibe.cnr.it)

**Abstract.**

A Pre–deployment calibration and a field validation of two low-cost (LC) stations equipped with O3 and NO2 metal oxide sensors were addressed. Pre–deployment calibration was performed after developing and implementing a comprehensive calibration framework including several supervised learning models, such as univariate linear and non–linear algorithms, as well

as multiple linear and non–linear algorithms. Univariate linear models included linear and robust regression, while univariate non–linear models included support vector machine, random forest, and gradient boosting. Multiple models consisted of both parametric and non-parametric algorithms. Internal temperature, relative humidity and gaseous interference compounds proved to be the most suitable predictors for multiple models, as they helped effectively mitigate the impact of environmental conditions and pollutant cross-sensitivity on sensor accuracy. A feature analysis, implementing Dominance analysis, feature

permutations and, SHapley Additive exPlanations method, was also performed to provide further insight into the role played by each individual predictor and its impact on sensor performances. This study demonstrated that while multiple random forest (MRF) returned higher accuracy than multiple linear regression (MLR), it did not accurately represent physical models beyond the Pre–deployment calibration dataset, so that a linear approach may overall be a more suitable solution. Furthermore, as well as being less computationally demanding and generally more suitable for non-experts, parametric models such as MLR have a

defined equation that also includes a few parameters, which allows easy adjustments for possible changes over time. Thus, drift correction or periodic automatable recalibration operations can be easily scheduled, which is particularly relevant for NO2 and O3 metal oxide sensors: as demonstrated in this study, they performed well with the same linear model form, but required unique parameter values due to inter-sensor variability.

## 1 Introduction

In the last few years, low–cost (LC) air quality sensors have gained increasing interest as they can provide near real–time observations with high spatial and temporal resolution. Their observations can be integrated into the current official regulatory





networks, usually monitoring air quality at lower space and time resolution, thus providing useful information to support policymakers and stakeholders in understanding air pollution dynamics (Brilli et al., 2021; Morawska et al., 2018). Nowadays, with advances in technology, LC sensors are able to measure pollutants such as particulate matter or gaseous species. Among
the latter, NO2 and O3 are the most commonly investigated since both short– and long–term exposure to these pollutants are associated with higher risk to human health (Ródenas García et al., 2022; World Health Organization, 2021).

Typically, LC NO2 and O3 monitors use electrochemical (EC) or metal oxide sensors (MOS) (Narayana et al., 2022; Concas et al., 2021; Idrees and Zheng, 2020), which produce an analog signal proportional to pollutant concentration.

In their simplest configuration, EC sensors are based on a redox reaction within an electrochemical cell in which the target
analyte oxidizes the anode or the cathode (Gäbel et al., 2022). As for MOS sensors, they have an exposed metal oxide surface film that changes its electrical properties when exposed to the target gas (Masson et al., 2015; Fine et al., 2010).

MOS sensors have a longer lifetime, can operate at higher temperatures and have a shorter response time and a wider operating range than EC sensors. By contrast, EC sensors have a lower power consumption as they do not require powering an electric heater, and are less impacted by high humidity levels (Narayana et al., 2022; Concas et al., 2021).
Overall, to choose between MOS and EC sensors depends on the goals of the deployment. EC sensors should be preferred in areas with steady temperatures and weather conditions (Concas et al., 2021), while MOS sensors are more suited for long-term monitoring (Concas et al., 2021; Narayana et al., 2022; Burgués and Marco, 2018). LC sensors are affected by environmental factors such as air temperature and relative humidity (Barcelo-Ordinas et al., 2019; Mueller et al., 2017; Mead et al., 2013) and suffer from cross-sensitivity with other air pollutants (Rai et al., 2017; Bart et al., 2014), thus complicating robust measurement
recovery. These issues depend on sensor characteristics such as the type of electrolyte, electrode, or semiconductor material used (Spinelle et al., 2015). Unfortunately, the lack of information or inconsistency in data sheets from sensor manufacturers makes it challenging to accurately interpret the readings (Narayana et al., 2022). As a result, these issues must be addressed in the calibration process to ensure accuracy and reliability of LC field measurements.

Although there is currently no established protocol for calibration (Karagulian, 2023), two main approaches to calibrating
LC sensors exist: Pre–deployment and field calibration. Pre–deployment calibration is typically performed in a controlled environment where LC sensors are exposed to a gas of known concentration in order to properly tune a calibration model (e.g., Claveau et al., 2022; Wei et al., 2018). Field calibration, on the other hand, consists in co-locating LC sensors near reference (official) stations that provide measured concentrations so as to develop a calibration model in real-world conditions (e.g., Spinelle et al., 2015). However, this approach may lead to potential inaccuracies when the calibrated LC sensors are deployed
on locations with varying air compositions and weather conditions (e.g., Spinelle et al., 2017; Aleixandre et al., 2013).

Both Pre–deployment and field calibration models are developed using a variety of mathematical methods, ranging from simple univariate regression models to more advanced machine learning techniques (Aula et al., 2022). The latter include various supervised learning techniques such as artificial neural networks (ANNs), random forest (RF), and support vector regression (SVR) (Karagulian, 2023; Karagulian et al., 2019; Cordero et al., 2018). In addition, the use of covariates such as
temperature and relative humidity, as well as interfering gasses as NO2, NO, and O3, can increase accuracy in the calibration process (Concas et al., 2021; Peterson et al., 2017; Piedrahita et al., 2014) . To date, while accuracy of LC calibration algorithms





has been widely investigated, there is a lack of studies addressing crucial issues associated to these techniques, such as: (i) transferability of field calibration beyond the training range (as highlighted Nowack et al. (2021); Zauli-Sajani et al. (2021); De Vito et al. (2020); Esposito et al. (2018)); (ii) Pre–deployment calibration complemented by a later field validation for EC

and MOS sensors (as mentioned in Maag et al. (2018)); (iii) the weight or importance of each feature included in multiple calibration models, particularly for black box techniques that cannot rely on statistical inference techniques (as discussed in Sahu et al. (2021)).

This study aims at addressing these issues by: (i) implementing a Pre–deployment calibration procedure for two LC stations measuring NO2 and O3 concentrations; (ii) identifying the optimal calibration that results in the highest accuracy; (iii)

performing a long-term (more than 1-year) field validation against a regulatory station located in a different site; (iv) critically discussing transferability and scalability of the selected calibration model for multiple devices. These goals have been pursued by using ten among parametric, non-parametric univariate and multiple algorithms. As a novel approach, the internal temperature of LC sensors was included in the covariate set for the multiple models, along with other important factors such as humidity and gaseous interference compounds. Furthermore, the study implemented model agnostic techniques such as

SHapley Additive exPlanations (SHAP, Lundberg and Lee (2017)) in order to evaluate model's generalization ability in a field environment. While SHAP has been used within previous pollution-related studies (e.g., Vega García and Aznarte, 2020), to the best of Authors' knowledge, no study applied it to LC sensors.

## 2   Materials and Methods

### 2.1   AIRQino low-cost stations

The study focuses on two AIRQino LC air quality monitoring stations (hereinafter AQ) developed by the Institute for BioEconomy of the National Research Council of Italy (IBE–CNR) in Florence (Italy), namely AQ1 and AQ2, equipped with MOS sensors to measure O3 and NO2 concentrations (Zaldei et al., 2017; Di Lonardo et al., 2014). AQ consists of an Arduino Shield Compatible electronic board that integrates LC and high temporal resolution sensors (2–3 min data acquisition frequency) to monitor environmental parameters and atmospheric pollutants such as relative humidity, internal and external temperature, CO,

CO2, O3, NO2, VOC, PM2.5, PM10. As for the atmospheric pollutants examined in this study (NO2 and O3), their concentrations are collected by SGX Sensortec MOS sensors: MiCS–2714 for NO2 (Sensortech, a) and MiCS–2614 for O3(Sensortech, b). These sensors consist of a micro metal oxide semiconductor diaphragm, with an integrated heating resistor (temperature ranges from 350 °C to 550 °C). The resistor-produced heat catalyses the reaction, which in turn affects the electrical resistance of the oxide layer itself. After the initial pre-heating period, the sensor detects gas changes in time intervals below 2 seconds.

The output signal from the sensor is passed through an analog–to–digital converter (ADC) circuit with a 10 bit output. The ADC converts the analog signal to a digital value between 0 and 1023 counts. This signal in counts is the primary output provided by the sensors (raw data). External air temperature (extT) and relative humidity (RH) are measured by an AM2305 (Asair) sensor protruding from the device enclosure. Internal temperature (intT) of the enclosure is monitored by a DS18B20



sensor (Maxim Integrated) that is mounted directly on the electronic board . Sensors' readings are collected by the onboard
microprocessor and sent to a PostgreSQL database via a general packet radio service (GPRS) connection.

## 2.2 Reference instruments

During Pre–deployment calibration, reference pollutant concentrations were measured using two HORIBA instruments (HORIBA
Ltd, Ambient Air Pollution AP SERIES analyzers). HORIBA model APNA–370 is an ambient nitrogen oxide monitor based
on the chemiluminescence principle, allowing a continuous measurement of NO and $NO_2$ concentrations. HORIBA model
APOA–370 was used to collect $O_3$ concentrations based on a cross flow modulated ultraviolet absorption method (Figure 1).

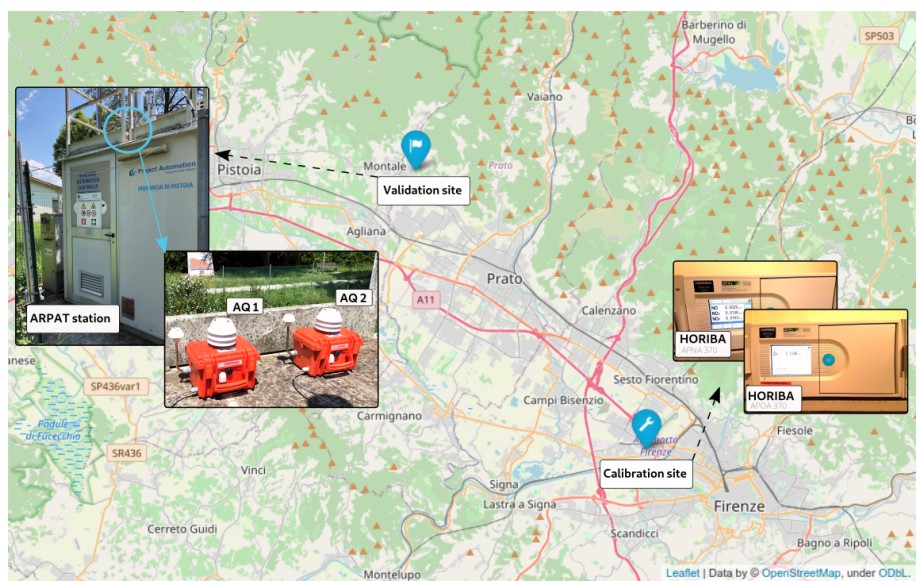

**Figure 1.** Map highlighting the calibration and validation locations for AQ1 and AQ2 LC air quality monitoring stations in Tuscany, Italy. At
the calibration site (Florence), the HORIBA instruments used for calibrating the LC stations are shown, while at the validation site (Montale),
the LC stations are pictured as installed on the roof of the reference ARPAT station (Air Quality Station EoI Code : IT1553A).

## 2.3 Sensor calibration

As detailed in Table S1 of the Supplementary material, Pre–deployment calibration of AQ1 and AQ2 stations against HORIBA
analyzers was performed at CNR-IBE headquarters in Florence, Italy (43°47'52" N, 11°11' E, Figure 1). HORIBA returned
measurements at 3 min resolution collected across a 70 day period (19 July 2017–27 September 2017). To ensure data validity,
measurements associated with RH>99 % following Wang et al. (2010) or classified as outliers by an interquartile range (IQR)
method (Dekking et al., 2005) were removed from the dataset, eventually resulting in 58949 valid records for $NO_2$, and 59261
valid records for $O_3$ concentrations. The workflow of the Pre–deployment calibration process is shown in Figure 2.



Prior to implementing the calibration techniques, an exploratory data analysis (EDA) was performed using the correlation matrix to identify important insights. Moreover, it is important to note that correlation does not always indicate causation. To investigate the connection between MOS sensors and temperature in relation to O3 (as described in Spinelle et al. (2016)), a K–means cluster analysis was conducted (MacQueen, J, 1965). To identify the optimal k number of clusters, the elbow method of the distortion metrics was employed (Bengfort et al., 2018).

The core of the calibration framework consisted of a set of supervised learning algorithms previously evaluated in the literature, falling in two categories: univariate and multiple models. The former are based on a single predictor (pollutant raw data), while the latter include additional predictors. Both categories included linear and non–linear algorithms. During the training phase, the datasets containing both LC and reference measurements were divided into a training subset consisting of 67 % of the data and a testing subset consisting of the remaining 33 %.

The suite of algorithms for univariate calibration linear methods included linear regression (Mijling et al., 2018; Maag et al., 2016) and robust regressions (Cavaliere et al., 2018) while the non–linear approaches comprised support vector machine (Bigi et al., 2018; Gu et al., 2018), random forest (Han et al., 2021; Zimmerman et al., 2018) and gradient boosting (Lin et al., 2018; Johnson et al., 2018). Multiple models, which considered temperature, humidity, and cross-sensitivity parameters for prediction, consisted of both parametric models and non-parametric models (Gäbel et al., 2022; Sayahi et al., 2020; Spinelle et al., 2017).

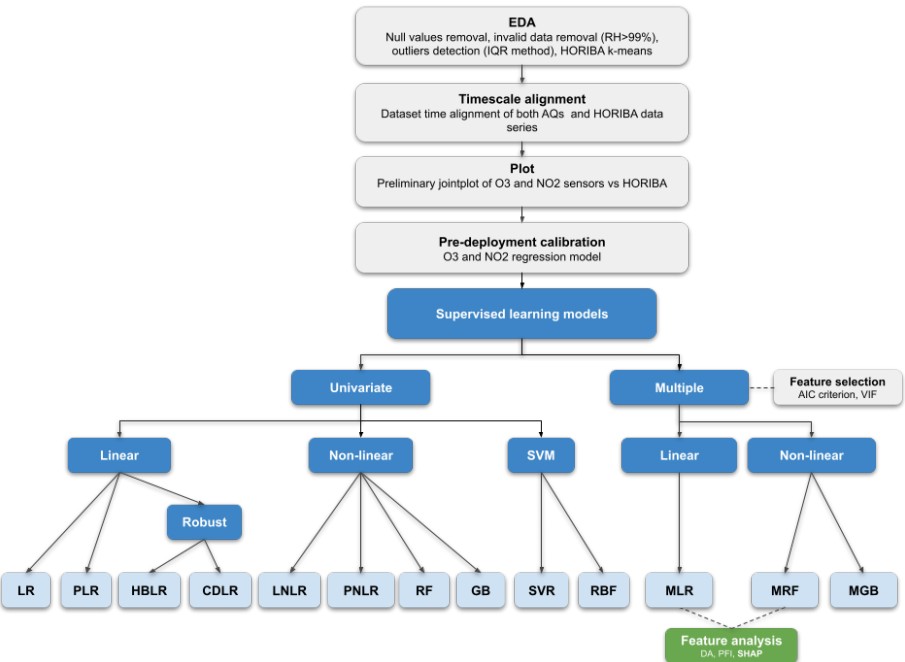

**Figure 2.** Workflow of the Pre–deployment calibration process performed in the work. The model abbreviations are also listed in the Appendix as Table A1.





### 2.3.1 Univariate models

The suite of univariate algorithms included a total of ten models, falling in three main categories: (i) linear regression (SLR), (ii) non–linear regression (SNLR), and (iii) support vector machine (SVM). Five regression models are included in SLR: simple linear regression (LR); polynomial regression of second (PLR2) and third (PLR3) degree; Huber regression (HBLR), a robust regression technique to outliers that uses a different loss function rather than the traditional least-squares; and Cook's distance regression (CDLR, Cook (1977)) which summarizes how much all values in the regression model change when the i-th obser-

vation is removed. SNLR included parametric and non-parametric models. The former included power non–linear regression (PNLR) and logarithm regression (LNLR), which considers the estimation of coefficients through the Levenberg–Marquardt algorithm. The latter included Random Forest (RF) and Gradient Boosting (GB). RF conducts optimal splitting of data samples into smaller sample sets, which then are fitted respectively along the tree paths, while GB built an additive model based on gradient boosting decision trees and in each stage a regression tree was fit. In the present calibration, the RF model used the

mean square error as a fitting function in order to evaluate each decision split and was configured with 100 decision trees, while the GB model used the squared error losses with 100 boosting stages. Finally SVM included support vector regression using linear kernel (SVR) and radial basis function (RBF). In SVM, the kernel allows to identify a hyperplane with maximum margin such that the maximum number of data points are within that margin. For each non–parametric models the grid search method was used to optimize the default hyper parameter values (Pedregosa et al., 2011; Smets et al., 2007).

### 2.3.2 Multiple models

Multiple models included both linear (MLR) and non–linear models, the latter consisting of multiple random forest (MRF) and multiple gradient boosting (MGB). While implementing an MLR model, a linear stepwise multi-regression analysis was carried out by automatically generating all possible models starting from a list of explanatory variables. In the case of NO2 and O3 sensors, the latter included intT, extT and RH. In order to solely include statistically significant variables, thus excluding

possible collinearity between them, the variance inflation factors (VIFs) were examined for each generated model. To refine the choice between intT and extT, a multiple linear model was used that alternatively incorporated both temperatures, followed by a cross-validation. Once a subset of significant explanatory variables was identified during MLR implementation, the MRF and MGB models were also applied: MGB was selected as GB is the univariate model that improves the results obtained by the supervised machine learning model, while MRF was selected as being a model widely used in the literature (e.g.,

Bisignano et al., 2022; Bigi et al., 2018; Zimmerman et al., 2018). When running the MRF model, all explanatory variables were considered and – as done while running the univariate RF model – the number of trees was 100 and the max depth of each tree was set to 10. Similarly, when running the MGB model, all explanatory variables were considered and – as done while running the univariate GB model – the number of boosting stages was 100. To compare the performance between models, specified metrics were evaluated such as the adjusted R-squared (AdjR$^2$, Draper and Smith (1998)).



### 2.3.3 Multiple models interpretation

To gain a better understanding of the impact due to different predictors and an insightful interpretation of the multiple model results, several analysis techniques have been applied, such as permutation feature importance (PFI, Breiman (2001)), dominance analysis (DA, Azen and Budescu (2003)), and SHapley Additive exPlanations (SHAP, Lundberg and Lee (2017)) analysis.

PFI is a model inspection technique that measures the global variable importance by observing the effect of randomly shuffling each explanatory variable. DA is a common procedure for identifying the relative importance of predictors in a linear model. In this work, five different DA statistics were evaluated: (i) interactional dominance (IntD); (ii) individual dominance (ID); (iii) average partial dominance (APD); (iv) total dominance (TD); (v) percentage relative importance (PRI).

SHAP analysis is a model-agnostic approach based on the game theory that can be applied to any machine learning model as a post hoc interpretation technique. According to the SHAP analysis, each machine learning model's prediction, $f(x)$, can be represented as the sum of its computed SHAP values, plus a fixed base value, as shown in Eq. (1):

$$f(x) = \Phi_0 + \sum_{i=1}^{p} \Phi_i \tag{1}$$

where $\Phi_0$ is the base value of the model, which represents the average prediction across all inputs, and $\Phi_i$ is the SHAP value for feature $i$ for the input $x$. Each $\Phi_i$ is computed as Eq. (2):

$$\Phi_i = \sum_{S \subseteq 1,2,...,p \setminus i} \frac{(p - |S| - 1)! \cdot |S|!}{p!} \cdot [f(x_{S \cup i}) - f(x_S)] \tag{2}$$

where $p$ is the total number of features, $S$ is a subset of all features except for feature $i$, $|S|$ is the number of features in subset $S$, $f(x_S)$ is the model's prediction for input $x$ with features in subset $S$, and $f(x_{S \cup i})$ is the model's prediction for input $x$ with features in subset $S$ and feature $i$ included.

SHAP values are calculated for each feature and value present in the dataset, and they approximate the contribution towards the output given by that data point. To compute SHAP values for different types of machine learning models, various SHAP implementations are available. In this study, the SHAP Linear Explainer function was used for MLR predictors, while the FastTreeSHAP explainer (Yang, 2021) was used for other models. Compared to the widely used TreeSHAP algorithm, FastTreeSHAP provides faster computation of feature importance values for tree-based models.

### 2.4 Field validation

To test Pre–deployment calibration models, the AQ stations were subject to a field validation based on hourly measurements collected during 429 consecutive days (19 June 2018–22 August 2019) by a reference air quality station operated by the Tuscany Region Environmental Protection Agency (ARPAT). High resolution NO2 and O3 concentrations measured by the AQ stations over the same period were hourly averaged in order to be aligned to the reference data . Overall, datasets of valid hourly records ranging 7383–9340 for NO2, and 7344–9303 for O3 concentrations, were used (Table S1). The reference air





quality station (EoI Code : IT1553A) was located at Montale, a small town in Tuscany located between the cities of Prato
and Pistoia (43°54'57" N, 11°00'26" E), and classified as a suburban background station (Figure 1). The ARPAT reference
station and the HORIBA APNA–370 analyzer used the same method for measuring NO2, while a different method (ultraviolet
photometry) was used by ARPAT to measure O3.

## 2.5    Statistics and libraries

The performances of each AQ station during both Pre–deployment calibration and field validation were computed using var-
ious statistical measures, including Pearson correlation coefficient (r), coefficient of determination ($R^2$), adjusted R–squared
(Adj$R^2$), root mean squared error (RMSE), normalized RMSE (nRMSE), which takes into account the range of values by
dividing the RMSE by the difference between the maximum and minimum values, mean absolute error (MAE), and mean
bias error (MBE). Variance impact factor (VIF) and Akaike information criterion (AIC) were also applied to discriminate
between MLR models. All calculations related to calibration procedure and analysis of performance of calibrated units are im-
plemented using Python Sklearn library (Pedregosa et al., 2011) and Python statsmodels module (Seabold and Perktold, 2010).
Finally, feature evaluation of MLR and MRF models was performed using python Dominance–Analysis library (Shekhar et al.,
2019), SHAP library (Lundberg and Lee, 2022), FastTreeSHAP library (Yang, 2022), and ELI5 Permutation Importance library
(TeamHG-Memex, 2022).

## 3    Results

### 3.1    Exploratory data analysis

After applying the humidity threshold and IQR procedure, 2 % and 12 % of records were withdrawn from the initial datasets
of AQ1 and AQ2 stations, respectively. The comparison between the resulting O3 and NO2 data and the HORIBA reference
concentrations is shown in Figure 3. Based on the analysis of Pearson's correlation (Fig. S2), three patterns for both AQ
stations emerged as conforming to the existing literature. HORIBA NO2 and O3 had a negative Pearson's r ($r_{AQ1}$=-0.77,
$r_{AQ2}$=-0.75), compatible with the chemical coupling of O3 and NOx=NO+NO2 (Han et al., 2011). AQ intT had a high
positive correlation with HORIBA O3 ($r_{AQ1}$=0.79, $r_{AQ2}$=0.80) compatible with the fact that high temperatures can increase
the rate of O3 formation through photochemical reactions (Han et al., 2011). AQ RH had a high negative correlation with
HORIBA O3 ($r_{AQ1}$=-0.75,$r_{AQ2}$=-0.74), compatible with the fact that high relative humidity is generally associated with lower
O3 levels (Camalier et al., 2007). Moreover, as a result of the convective heat transfer equation, a strong positive correlation was
observed between intT and extT for each AQ ($r_{AQ1,AQ2}$=1). On average, the temperature difference between intT and extT
remains relatively constant at around 8 °C. A visual representation of the difference between the two temperatures, plotted
against their mean, can be found in the Bland–Altman plots on Figure S3. No significant correlation was observed between
NO2 raw and either temperature or RH. Moderate positive associations were instead found between O3 raw and both intT
($r_{AQ1}$=0.55; $r_{AQ2}$=0.55) and extT ($r_{AQ1}$=0.52; $r_{AQ2}$=0.53). A k–means clustering analysis was then performed on HORIBA





O3 of both AQs, resulting in the identification of six distinct clusters for both AQs. Figure 4 showed O3 raw and internal temperature (intT) regression plot for each cluster, along with the corresponding regression formula.

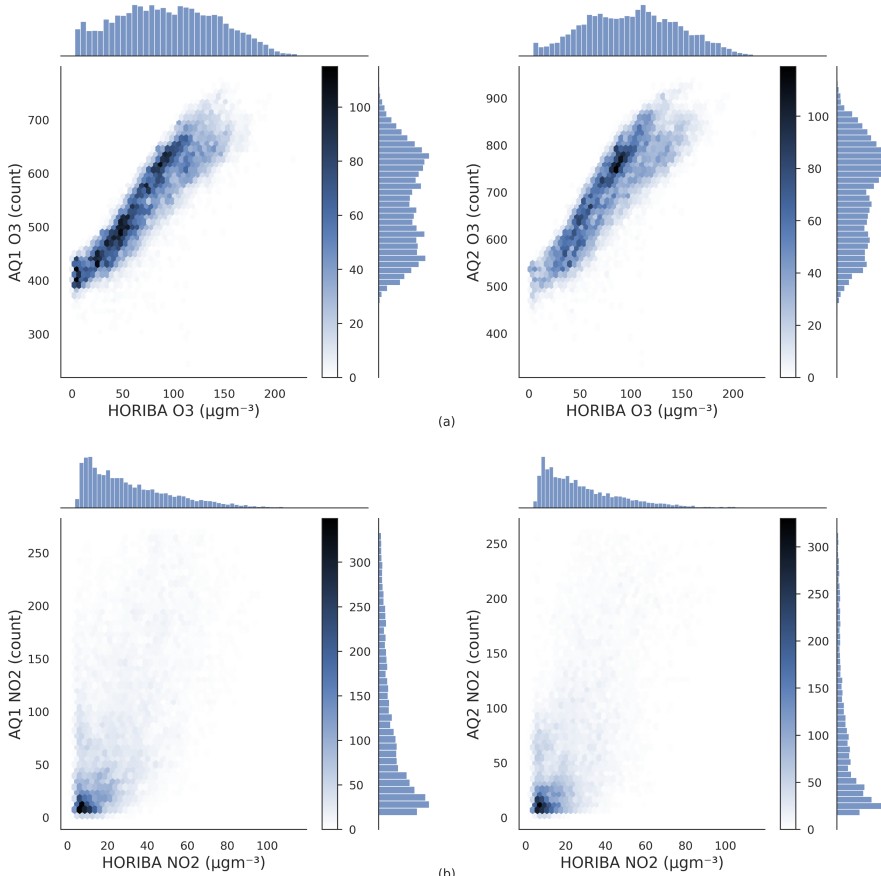

**Figure 3.** Joint Plots of 3 min sampled AQ1 and AQ2 signals vs. HORIBA reference concentrations observed during Pre–deployment calibration: O3(a); NO2(b). Data points are plotted into different hex bins and the counting are indicated with the colour bar.





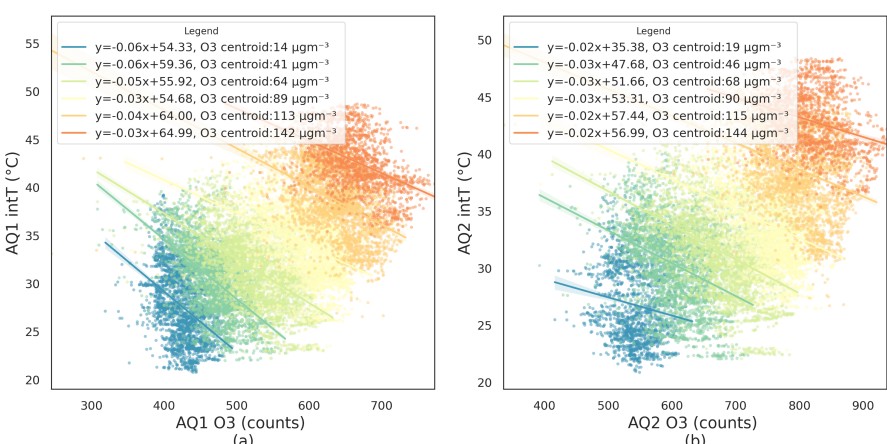

**Figure 4.** Relationship between O3 raw data and intT in K–means clusters for AQ1 (a) and AQ2 (b) stations. Regression lines were fitted to each cluster. The coefficients for each regression line and the centroid of each cluster are reported in the legend.




## 3.2 Univariate models

The results of the supervised linear (SLR), supervised non–linear (SNLR) and support vector machine (SVM) models applied for both AQ stations and pollutants are reported in Table 1. For both AQ stations, the best performances were found using the 215 GB model, with O3 concentrations generally better fitted than NO2 concentrations.

**Table 1.** Table 1. Statistics of the univariate regression models applied to the AQ1 and AQ2 stations. Note that for non–linear models (LNLR and PNLR) $R^2$ is not a useful metric, while it is for linear models that use polynomials to model curvature in the data (Spiess and Neumeyer, 2010)

| Pollutant | AQ id | Stat. | SLR | | | | | SNLR | | | | SVM | |
|---|---|---|---|---|---|---|---|---|---|---|---|---|---|
| | | | LR | PLR2 | PLR3 | CDLR | HBLR | LNLR | PNLR | RF | GB | SVR | RBF |
| O3 | AQ1 | $R^2$ | 0.81 | 0.81 | 0.82 | 0.81 | 0.81 | 0.00 | - | 0.82 | 0.82 | 0.81 | 0.70 |
| | | RMSE | 16.92 | 16.92 | 16.75 | 16.99 | 17.00 | 17.15 | 18.85 | 16.78 | 16.58 | 17.18 | 21.31 |
| | | MAE | 13.42 | 13.41 | 13.40 | 13.19 | 13.18 | 13.47 | 14.94 | 13.42 | 13.27 | 13.12 | 16.06 |
| | | MBE | -0.28 | -0.28 | -0.30 | 1.56 | 1.69 | -0.33 | -0.96 | -0.18 | -0.20 | 2.89 | 3.53 |
| | AQ2 | $R^2$ | 0.77 | 0.77 | 0.77 | 0.76 | 0.77 | - | - | 0.77 | 0.78 | 0.76 | 0.60 |
| | | RMSE | 17.58 | 17.55 | 17.41 | 17.86 | 17.75 | 17.97 | 18.46 | 17.58 | 17.36 | 18.11 | 23.06 |
| | | MAE | 14.11 | 14.10 | 14.10 | 13.81 | 13.85 | 14.30 | 14.79 | 14.18 | 14.05 | 13.79 | 17.38 |
| | | MBE | 0.14 | 0.14 | 0.15 | 3.03 | 2.39 | 0.19 | -0.48 | 0.24 | 0.14 | 4.17 | 3.55 |
| NO2 | AQ1 | $R^2$ | 0.34 | 0.34 | 0.34 | 0.32 | 0.33 | - | - | 0.33 | 0.35 | 0.33 | 0.31 |
| | | RMSE | 14.22 | 14.18 | 14.16 | 14.40 | 14.28 | 14.51 | 14.46 | 14.35 | 14.14 | 14.35 | 14.50 |
| | | MAE | 10.91 | 10.84 | 10.82 | 10.77 | 10.79 | 11.20 | 11.22 | 10.84 | 10.75 | 10.76 | 10.81 |
| | | MBE | 0.09 | 0.11 | 0.11 | 2.26 | 1.21 | 0.16 | -0.11 | 0.06 | 0.09 | 1.73 | 2.08 |
| | AQ2 | $R^2$ | 0.38 | 0.38 | 0.38 | 0.35 | 0.37 | - | - | 0.36 | 0.38 | 0.36 | 0.32 |
| | | RMSE | 12.86 | 12.85 | 12.85 | 13.12 | 12.95 | 13.45 | 13.06 | 13.04 | 12.85 | 13.02 | 13.39 |
| | | MAE | 9.83 | 9.83 | 9.84 | 9.69 | 9.69 | 10.37 | 10.06 | 9.94 | 9.81 | 9.67 | 9.83 |
| | | MBE | -0.09 | -0.09 | -0.09 | 2.56 | 1.53 | -0.05 | -0.24 | -0.05 | -0.10 | 2.05 | 2.60 |

## 3.3 Multiple regression

EDA suggested that the inclusion in multiple regression models of both intT and extT may result in unstable results due to their strong collinearity ($r_{AQ1,AQ2}$=1). This was confirmed by the variance inflation factor (VIF) for the MLR model, which was higher than 5 when both variables were used (Table S4). To ensure consistent selection of the optimal temperature variable in 220 the model, cross-validation procedure was conducted on the calibration dataset for the MLR model, alternately including intT



and extT in the covariate set. The results of 5–split cross–validation (Table 2) showed no significant differences using intT or extT, whilst the use of intT provided a slightly higher mean accuracy and a lower mean RMSE.

**Table 2.** $R^2$ and RMSE ($\mu gm^{-3}$) values by covariate set including intT or extT variables of the cross–validation procedure applied to the MLR model.

| AQ id | Pollutant | Stat. | Covariate set (mean±SD) | |
|-------|-----------|-------|-------------------------|--|
| | | | O3,NO2,intT,RH | O3,NO2,extT,RH |
| AQ1 | O3 | $R^2$ | 0.93±0.03 | 0.93±0.02 |
| | | RMSE | 9.52±2.51 | 9.55±2.10 |
| AQ2 | O3 | $R^2$ | 0.91±0.04 | 0.91±0.05 |
| | | RMSE | 9.52±2.86 | 9.72±2.85 |
| AQ1 | NO2 | $R^2$ | 0.57±0.21 | 0.56±0.24 |
| | | RMSE | 10.75±1.31 | 10.83±1.43 |
| AQ2 | NO2 | $R^2$ | 0.61±0.07 | 0.61±0.07 |
| | | RMSE | 9.87±2.07 | 9.89±2.02 |

Following the previous result, the final subset of predictors used for all models consisted of intT, RH, and raw signal from both sensors (Tables S5 and S6). Accordingly, for both stations, Eq.3 and Eq.4 were the best model formulas for O3 and NO2 sensors, respectively:

$$O_3 = \beta_0 + \beta_1 \cdot NO_2raw + \beta_2 \cdot O_3raw + \beta_3 \cdot RH + \beta_4 \cdot intT \tag{3}$$

$$NO_2 = \beta_0 + \beta_1 \cdot NO_2raw + \beta_2 \cdot O_3raw + \beta_3 \cdot RH + \beta_4 \cdot intT \tag{4}$$

The calibration coefficients achieved for the MLR model are reported in Table 3, while the scores of MLR, MGB and MRF model application are reported in Table 4.

**Table 3.** Statistics of the MLR model applied to the AQ1 and AQ2 stations. $\beta_0$ are the intercepts and $\beta_i$ the calibration coefficients.

| Pollutant | AQ id | Coefficient | | | | | Stat. |
|-----------|-------|-------|-------|-------|-------|-------|-------|
| | | $\beta0$ | $\beta1$ | $\beta2$ | $\beta3$ | $\beta4$ | $AdjR^2$ |
| O3 | AQ1 | -180.76 | -0.11 | 0.23 | 0.15 | 3.79 | 0.95 |
| | AQ2 | -133.43 | -0.16 | 0.14 | 0.03 | 3.58 | 0.95 |
| NO2 | AQ1 | 144.78 | 0.05 | -0.14 | -0.32 | -0.93 | 0.69 |
| | AQ2 | 126.78 | 0.08 | -0.10 | -0.23 | -0.87 | 0.69 |





Overall, O3 concentrations were better fitted than NO2 concentrations, while MRF proved to be the finest model, generally outperforming MGB and particularly MLR model.

**Table 4.** Statistics of the multiple regression models applied to the AQ1 and AQ2 stations.

| Pollutant | AQ id | Stat. | Multiple models | | |
|---|---|---|---|---|---|
| | | | MLR | MGB | MRF |
| O3 | AQ1 | AdjR$^2$ | 0.95 | 0.97 | 0.98 |
| | | RMSE | 8.62 | 7.30 | 6.04 |
| | | MAE | 6.30 | 5.40 | 4.31 |
| | | MBE | -0.10 | -0.01 | -0.01 |
| | AQ2 | AdjR$^2$ | 0.95 | 0.96 | 0.98 |
| | | RMSE | 8.58 | 6.86 | 5.51 |
| | | MAE | 6.50 | 5.17 | 4.05 |
| | | MBE | -0.03 | -0.03 | 0.09 |
| NO2 | AQ1 | AdjR$^2$ | 0.69 | 0.80 | 0.86 |
| | | RMSE | 9.68 | 7.84 | 6.63 |
| | | MAE | 7.36 | 5.76 | 4.72 |
| | | MBE | -0.03 | 0.06 | 0.06 |
| | AQ2 | AdjR$^2$ | 0.69 | 0.80 | 0.85 |
| | | RMSE | 9.07 | 7.28 | 6.30 |
| | | MAE | 6.83 | 5.35 | 4.46 |
| | | MBE | -0.08 | 0.03 | 0.05 |



## 3.4 Multiple models interpretation

The DA statistics and PFI weights achieved for MLR and MRF models are reported in Table 5. Overall, the DA analysis showed that O3 raw and intT were the most important features for both stations and pollutants. In particular, for O3 concentrations O3 raw data resulted in the highest PRI value, explaining 38.96 % and 34.95 % of the $R^2$ of the MLR model for AQ1 and AQ2, respectively, followed by intT (28.64 % and 31.51 %, respectively). Also for NO2 concentrations, O3 raw data had the highest PRI value, explaining 55.18–51.13 % of the $R^2$ of the MLR model, followed by NO2 raw data (23.78–26.79 %). In O3 MRF regression, O3 raw was the most important feature for AQ1, while it was intT for AQ2. Conversely, in NO2 MRF regression, O3 raw was the most important feature for both AQ stations followed by RH for AQ1 and by NO2 for AQ2. Notably, for both MLR and MRF models, O3 raw proved to be a more important feature in NO2 calibration than in O3 calibration.

**Table 5.** DA statistics and PFI weights achieved for MLR and MRF models applied to the AQ1 and AQ2 stations.

| Pollutant | AQ id | Variable | IntD | ID | DA APD | TD | PRI | PFI Weight |
|-----------|-------|----------|------|------|------|------|-------|-----------------|
| O3 | AQ1 | O3 | 0.09 | 0.82 | 0.29 | 0.37 | 38.96 | 0.66 ± 0.01 |
| | | intT | 0.07 | 0.62 | 0.20 | 0.27 | 28.64 | 0.48 ± 0.01 |
| | | RH | 0.00 | 0.57 | 0.10 | 0.19 | 19.92 | 0.01 ± 0.00 |
| | | NO2 | 0.02 | 0.26 | 0.10 | 0.12 | 12.47 | 0.16 ± 0.01 |
| | AQ2 | O3 | 0.06 | 0.77 | 0.25 | 0.33 | 34.95 | 0.47 ± 0.01 |
| | | intT | 0.11 | 0.64 | 0.22 | 0.30 | 31.51 | 0.55 ± 0.01 |
| | | RH | 0.00 | 0.55 | 0.09 | 0.18 | 19.10 | 0.01 ± 0.00 |
| | | NO2 | 0.03 | 0.31 | 0.10 | 0.14 | 14.44 | 0.20 ± 0.01 |
| NO2 | AQ1 | O3 | 0.16 | 0.66 | 0.36 | 0.39 | 55.18 | 1.12 ± 0.02 |
| | | NO2 | 0.02 | 0.34 | 0.15 | 0.17 | 23.78 | 0.21 ± 0.01 |
| | | intT | 0.02 | 0.20 | 0.06 | 0.08 | 11.93 | 0.18 ± 0.01 |
| | | RH | 0.03 | 0.18 | 0.02 | 0.06 | 9.11 | 0.22 ± 0.01 |
| | AQ2 | O3 | 0.12 | 0.64 | 0.33 | 0.35 | 51.13 | 1.01 ± 0.04 |
| | | NO2 | 0.04 | 0.38 | 0.16 | 0.19 | 26.79 | 0.19 ± 0.01 |
| | | intT | 0.03 | 0.21 | 0.06 | 0.09 | 13.34 | 0.18 ± 0.01 |
| | | RH | 0.03 | 0.18 | 0.02 | 0.06 | 8.74 | 0.16 ± 0.01 |

A SHAP analysis was also performed in order to gain insight into both global and local contribution of each feature at both individual instance level and across the population, resulting in the SHAP bee swarm plots for MLR and MRF shown in Figures 5 and 6, respectively. The bee swarm plot ranks the input features from the highest to the lowest mean absolute SHAP values for the entire dataset. For each variable, every instance of the dataset appears as its own point. The points are distributed



horizontally along the x-axis according to their SHAP value. In places where there is a high density of SHAP values, the points are stacked vertically. The color bar corresponds to the raw values of each feature for each instance, providing a visual representation of the feature's contribution to the outcome prediction.

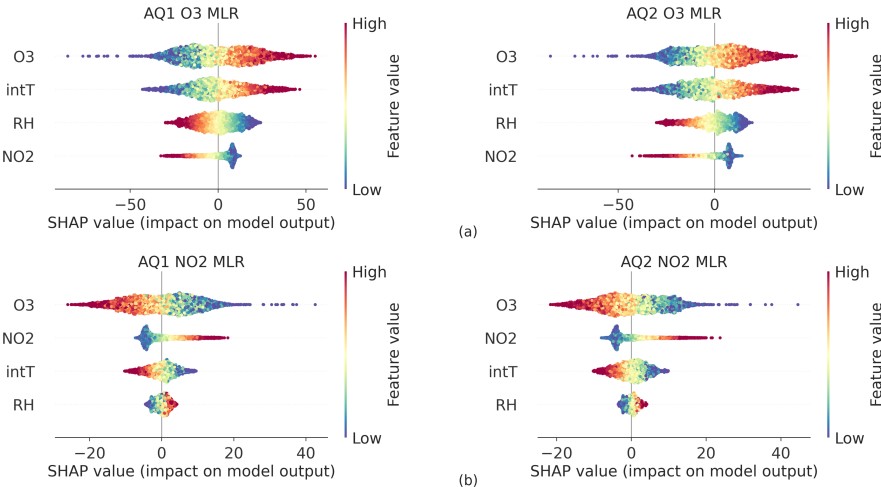

**Figure 5.** Bee swarm plot showing the SHAP values calculated for each feature and instance using the linear explainer the MLR model for O3 (a) and NO2 (b).

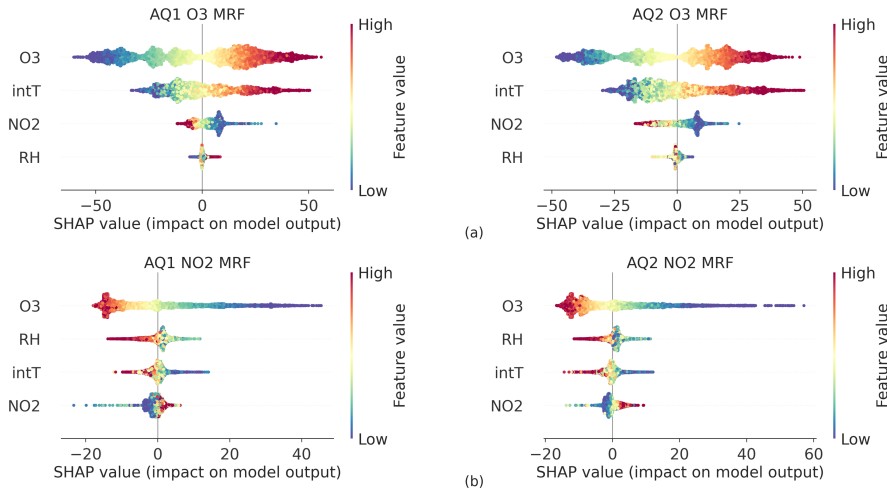

**Figure 6.** Bee swarm plot showing the SHAP values calculated for each feature and instance using the fast tree explainer of the MRF model for O3 (a) and NO2 (b).



As for the MLR model for both AQ stations, high levels of both O3 raw and intT data had a strong and positive impact on O3 output, as indicated by high and positive SHAP values (Fig. 5a), while high levels of NO2 raw data had a strong and positive

impact on NO2 output (Figure 5b). Herein, however, high levels of O3 raw and intT data had a greater impact on decreasing the predicted values of NO2 than the raw data of NO2.

Also for the MRF model O3 raw data had a high influence on O3 predicted values (Fig. 6a): higher values of O3 raw data increased O3 prediction, while lower values had a negative effect. This also applies to the NO2 output values (Fig. 6b), as higher values of O3 raw data decreased NO2 prediction and lower values had a positive effect. Herein, however, high or low

levels of NO2 raw had no significant influence on the prediction.The mean absolute SHAP values for all features of both MLR and MRF models are reported in Figure S7.

In order to provide a local interpretability, a heatmap for the SHAP values of the NO2 MLR model was also elaborated (Figure 7). The heatmap showed that lower model predictions $f(x)$, computed using Eq.(1), were linked to an absolute blue colour for O3 and a light blue colour for NO2 for both AQs. This suggested that O3 raw data had a more significant impact

mostly on the lower NO2 concentrations than NO2 raw itself had while the impact of NO2 raw data became significant at higher concentrations.

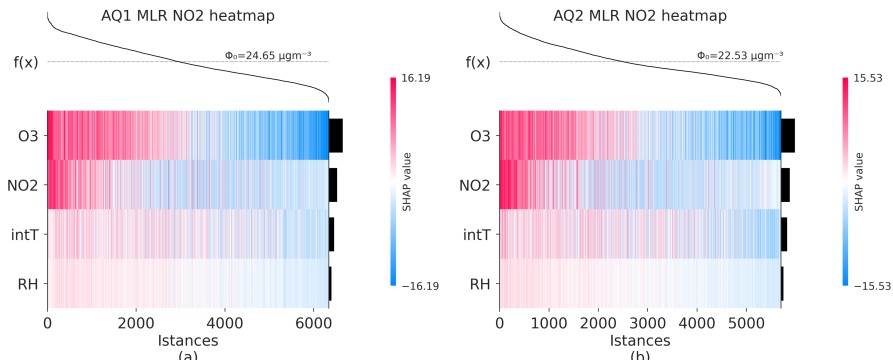

**Figure 7.** Heatmap of SHAP values of the NO2 MLR model for AQ1 (a) and AQ2 (b). The heatmap displays the contribution of each feature to the model's predictions, with positive contributions represented by red cells and negative contributions by blue cells. Colour intensity denotes the magnitude of the contribution. The output of the model, $f(x)$, is shown above the heatmap matrix, centred around the explanation's base value ($\phi_0$), and the global importance of each model input is shown in the bar plot on the right–hand side of the plot. Observations have been ordered by the sum of the SHAP values over all features.



## 3.5   Field validation

The scores of field validation involving the MLR and MRF calibrated models are summarized by the Taylor diagram (Figure 8). Model accuracy in predicting O3 concentrations (Fig. 8a) is confirmed to be higher than in predicting NO2 concentrations (Fig. 8b). In terms of Pearson's r values, the MLR model outperforms the MRF model, as exhibiting r values (0.92–0.93 for O3 and 0.75–0.78 for NO2) higher than MRF (0.81–0.76 for O3 and 0.76–0.65 for NO2), while the opposite applies in terms of standard deviation, as MRF returns lower values than MLR. Notably, a significant difference by AQ station may be observed in MRF scores, while it is not the case for MLR. The detailed statistics of MLR and MRF field validation may be found in Table S9, while weekly concentrations predicted by the models against the reference station are given in Figure S10.

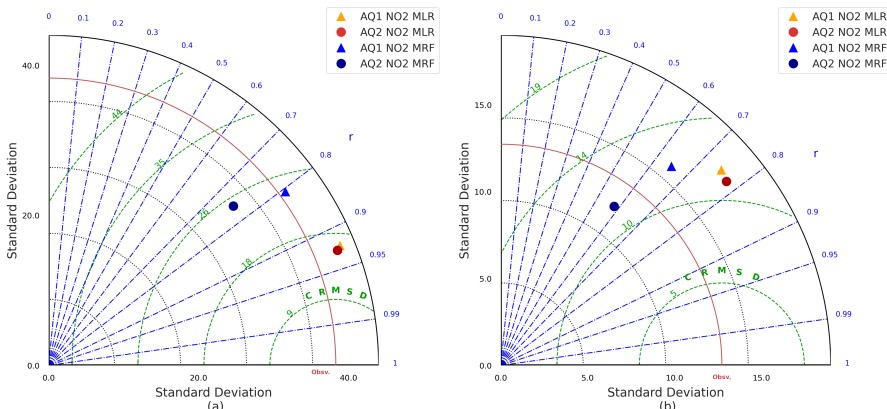

**Figure 8.** Taylor diagrams of the pre–deployment MLR and MRF calibrated models assessed against the ARPAT reference station for O3 (a) and NO2 (b) concentrations. The Taylor diagram (Rochford, 2016) consists of a polar plot in which the radial distance from the origin represents the standard deviation of predictions, and the angle represents the correlation between predictions and observations. Models that agree well with observations lie closer to the red line. CRMSD is a relative measure of model fit, providing a normalized measure of the deviation from the actual values. This deviation is normalized by the standard deviation of the reference values, which allows a fair comparison of models' performance.

A seasonal analysis was also performed for MLR field validation (Table 6). O3 concentrations were well predicted across all seasons: for both stations, the lowest nRMSE values were registered in the summers 2018 and 2019, and the highest in winter 2018–2019. Notably, all statistical scores during the summer 2019 proved to be worse compared to the summer 2018, suggesting a likely drift in sensor accuracy after one year of deployment. As for NO2, the highest (and thus more meaningful) concentrations were measured in winter 2018–2019. The NO2 scores during this period, however, confirm to be worse than those affecting O3 during the period of highest O3 concentrations (i.e. summers 2018 and 2019). Furthermore, the scores of seasonal analysis addressed for MRF field validation may be found in Table S11.





**Table 6.** Seasonal analysis of MLR validation. Min–Max ($\mu gm^{-3}$) represent the minimum and maximum concentrations measured by the reference station, while intT (°C) the average internal temperature measured by the AQ stations.

| Year | Season | Pollutant | AQ id | min–max | intT | r | nRMSE | MAE | MBE |
|------|--------|-----------|-------|---------|------|---|-------|-----|-----|
| | | | | | | | Stat. | | |
| 2018 | Summer | O3 | AQ1 | 6–166 | 34.65 | 0.94 | 9.17 | 11.69 | -5.17 |
| | | | AQ2 | 6–166 | 34.20 | 0.94 | 8.80 | 11.07 | 7.57 |
| | | NO2 | AQ1 | 1–47 | 34.62 | 0.69 | 35.74 | 14.04 | 13.83 |
| | | | AQ2 | 1–47 | 34.16 | 0.69 | 16.97 | 5.94 | 2.90 |
| | Autumn | O3 | AQ1 | 2–146 | 28.06 | 0.93 | 13.24 | 15.08 | 11.35 |
| | | | AQ2 | 2–146 | 25.53 | 0.94 | 15.87 | 19.32 | 18.37 |
| | | NO2 | AQ1 | 1–62 | 28.07 | 0.73 | 19.66 | 8.62 | 5.57 |
| | | | AQ2 | 1–62 | 25.54 | 0.71 | 16.57 | 7.60 | -2.40 |
| | Winter | O3 | AQ1 | 2–65 | 16.60 | 0.93 | 17.94 | 8.07 | 0.11 |
| | | | AQ2 | 2–72 | 14.53 | 0.93 | 18.97 | 8.97 | 2.50 |
| | | NO2 | AQ1 | 3–88 | 16.59 | 0.71 | 21.01 | 13.97 | 10.44 |
| | | | AQ2 | 2–88 | 14.53 | 0.70 | 18.34 | 12.16 | 4.62 |
| 2019 | Spring | O3 | AQ1 | 2–132 | 22.41 | 0.86 | 13.32 | 13.98 | 3.36 |
| | | | AQ2 | 2–132 | 21.14 | 0.84 | 13.91 | 14.40 | 4.63 |
| | | NO2 | AQ1 | 2–63 | 22.32 | 0.74 | 13.18 | 6.10 | -2.09 |
| | | | AQ2 | 2–63 | 21.09 | 0.67 | 16.19 | 7.31 | -5.36 |
| | Summer | O3 | AQ1 | 7–185 | 34.98 | 0.92 | 10.09 | 14.62 | 10.21 |
| | | | AQ2 | 7–185 | 32.29 | 0.92 | 10.68 | 15.98 | 12.71 |
| | | NO2 | AQ1 | 0–47 | 34.94 | 0.63 | 17.82 | 6.36 | 3.58 |
| | | | AQ2 | 0–47 | 32.25 | 0.68 | 13.68 | 5.01 | -3.05 |



## 4   Discussion

Current outcomes achieved for MOS O3 and NO2 Pre–deployment calibration are generally consistent with those found in the literature. Agreeing with Nowack et al. (2021), for example, MOS NO2 calibration has low accuracy in linear univariate

models. As also reported by Spinelle et al. (2015) using a simple linear field calibration, a poor $R^2$ (0.21) was achieved for the MiCS–2710 NO2 sensor as compared to the value (0.845) achieved for the O3_3E1F EC sensor. Within a field calibration performed in Barcelo-Ordinas et al. (2019) using a simple LR model, RMSE=9.45 $\pm$ 2.74 ppb was achieved for NO2 and RMSE=4.9 $\pm$ 1.68 ppb for O3.

     As shown in Table 1, O3 sensor calibration returned quite high $R^2$ values, suggesting a limited potential room for improve-

ment using more complex univariate techniques such as SVR, RF or GB, as previously noted in Sales-Lérida et al. (2021). The opposite applies to NO2 calibration, whose quite low $R^2$ values urged to implement more sophisticated models primarily incorporating multiple covariates. The importance of including multiple covariates such as temperature, humidity, and gaseous interference compounds resulting in better performances for both EC and MOS sensors was emphasized within several studies cited in Karagulian et al. (2019). For example, incorporating temperature and relative humidity into an MLR model improves

the calibration of NO2 sensors, resulting in $R^2$ values ranging from 0.6 to 0.9 (Mijling et al., 2018). A further improvement can be achieved by including O3 as a predictor, resulting in $R^2$ generally above 0.9 (Karagulian et al., 2019). Moreover, several studies (e.g., Bisignano et al., 2022; Johnson et al., 2018) have demonstrated that also "black box" machine learning models such as MRF or MGB can effectively calibrate LC sensors and mitigate the impact of environmental conditions and pollutant cross–sensitivity.

It is confirmed that both linear and non–linear multiple models result in a slight improvement in O3 prediction and a significant one in NO2 prediction compared to univariate models (Table 4). In particular, the MLR model improved the accuracy of the simple LR by more than 14–18 % for O3 and 35–31 % for NO2. The MRF model proved to be the most effective among the tree–based models. In addition, the study introduced a novel approach by using internal rather than external temperature to account for the effect of temperature on sensor readings and tackle possible issues in the board's analog–to–digital converter

circuit. This approach was supported by the slight $R^2$ increase (Table 2), and by the k–means cluster plots (Figure 4). The latter revealed a consistent and comparable correlation between intT and O3 across all clusters, marked by a generally stable slope in the correlation lines. These results – also supported by the correlation matrices (Fig. S1) – provided further evidence of usefulness in including intT as an explanatory variable in order to better understand the O3 sensor patterns. However, the moderate correlation between temperature and reference O3 raised concerns about potential weight biases resulting when esti-

mating O3 concentration using a multiple approach, especially during the summer–winter cycle. To gain different perspectives on the model, three feature importance measures were conducted for both MLR and MRF models, enabling the evaluation of the models from a "true to the data" and a "true to the model" perspective. In terms of traditional statistical inference techniques such as DA and PFI during MLR and MRF O3 calibration, the results primarily confirmed O3 raw and secondly intT as the most significant predictors (Table 5), which were consistent with those reported by Masson et al. (2015). For NO2 calibration,

the O3 raw data had the highest rank, indicating a significant cross–sensitivity effect. The challenge with traditional feature




selection methods like DA and PFI is that they may produce misleading results when features are highly correlated or the data is noisy. These methods in fact do not consider interactions or correlations between predictors, and DA is only applicable to linear models. To overcome these limitations, the study utilized SHAP analysis. SHAP heatmap (Fig. 7) revealed the O3 sensor's ability to cope with the lower reading limit of the NO2 sensor for both AQs. This is particularly important at low NO2 concentrations, where the NO2 sensor might not provide accurate readings. In such cases, the O3 sensor can help to fill in the gaps and provide more complete data.

Assessed during a quite long (more than 1 year) field validation period, the calibrated MLR and MRF models proved to be able to capture the variations of O3 and NO2 measured concentrations, although with a different accuracy (Figure 8). Both models exhibited a clear decline in performances over time, with O3 models, however, resulting less affected than NO2 models, which confirms findings from Sahu et al. (2021). Notably, there was a minimal decline in performance observed during the winter period, despite the Pre–deployment calibration being mainly conducted during the summer.

Noteworthy, although achieving better results during the Pre–deployment calibration, the MRF model was outperformed by the MLR during the field validation, which confirmed the limitations of the black box models, previously detected in the literature (e.g., Nowack et al., 2021; Malings et al., 2019), in extrapolating and predicting pollution levels beyond the calibration dataset. Thus, SHAP analysis of the MRF model indicates that it was not "true" to the expected underlying physical model.

## 5 Conclusions and Perspective

In this study, Pre–deployment calibration and field validation of two low–cost (LC) stations named "AIRQino", developed by IBE–CNR in Florence (Italy), were addressed. The stations were equipped with O3 and NO2 MOS sensors, as well as meteorological sensors. Pre–deployment calibration was performed after developing and implementing a comprehensive calibration framework, consisting of several among parametric, non–parametric univariate and multiple algorithms, that allowed to identify the optimal calibration pathway. With an eye on assessing whether sacrificing model interpretability for greater calibration accuracy was worthwhile, it was eventually demonstrated robust LC performances outside the training conditions and capability of easy adjustments due to changes in sensor performances over time. While selecting the most suitable LC calibration models, necessarily going beyond mere accuracy, this study primarily recommends to: (i) include multiple covariates such as internal (rather than external) temperature, relative humidity and gaseous interference compounds into multiple regression models; (ii) analyse the importance of the features used in the multiple models, so as to disclose their role when the calibrated LC stations should be operated under field conditions (rather than in a controlled environment). As a novelty applied to LC sensor calibration, the SHapley Additive exPlanations (SHAP) method was used to provide further insight into the role played by model individual predictors and their global and local impact on the overall LC sensor performances. This method was also used to hypothesize the model's capability to accurately describe conditions beyond the Pre–deployment calibration. This study confirmed that machine learning models such as MRF can effectively calibrate LC sensors and mitigate the impact of environmental conditions and pollutant cross–sensitivity. However, while the MRF model returned higher accuracy than MLR, it did not accurately represent physical models beyond the Pre–deployment calibration dataset, so that a linear approach may



overall be a more suitable solution. Furthermore, as well as being less computationally demanding and generally more suitable
for non–experts, parametric models such as MLR have a defined equation that also includes a few parameters, which allows –
when needed – easy adjustments for possible changes over time. Thus, drift correction or periodic automatable recalibration
operations can be easily scheduled. This is particularly relevant for NO2 and O3 MOS sensors: as demonstrated in this study,
they performed well with the same linear model form, but required unique parameter values due to inter–sensor variability.

A limitation of the present work is that the LC stations have been calibrated during a period not particularly long (70
days) and a typically summer one, thus when pollution levels are generally meaningful for O3, but they are not for NO2
concentrations. To this aim, in the future a Pre–deployment calibration during a winter period where NO2 concentrations
are generally higher should be performed. As well as testing real–world consistency, stability, and durability of the AIRQino
LC stations, this should help improve NO2 sensor performances. Furthermore, based on the results obtained from the SHAP
analysis and taking into account the good calibration results achieved for black box models, a possible future modelling step
could be to build a hybrid parametric and non parametric model, as proposed in various study (e.g., Si et al., 2020; Lin et al.,
2018; Zimmerman et al., 2018).



# Appendix A

**Table A1.** Nomenclature

| | | | | | |
|---|---|---|---|---|---|
| **APD** | Average partial dominance | **LC** | Low–cost | **RBF** | Radial basis function |
| **AQ** | AIRQino | **LNLR** | Logarithm regression | **RF** | Random Forest |
| **CDLR** | Cook's distance regression | **LR** | Linear regression | **RH** | Relative humidity |
| **DA** | Dominance Analysis | **MGB** | Multiple gradient boosting | **SHAP** | SHapley Additive exPlanations |
| **EC** | Electrochemical | **MLR** | Multiple linear regression | **SLR** | Supervised linear regression |
| **EDA** | Exploratory data analysis | **MOS** | Metal oxide sensors | **SNLR** | Supervised nonlinear regression |
| **extT** | External Temperature | **MRF** | Multiple random forest | **SVM** | Support vector machine |
| **GB** | Gradient Boosting | **PFI** | Permutation feature importance | **SVR** | Support vector regression |
| **HBLR** | Huber regression | **PLR2** | Polynomial regression of second degree | **TD** | Total dominance |
| **ID** | Individual dominance | **PLR3** | Polynomial regression of third degree | **VIF** | Variance impact factor |
| **IntD** | Interactional dominance | **PNLR** | Power nonlinear regression | | |
| **intT** | Internal temperature | **PRI** | Percentage relative importance | | |

*Code and data availability.* All data (HORIBA reference data, AIRQinos raw signal data , ARPAT validation data), and codes (Jupyter notebook) to recreate the results discussed here are provided online at https://doi.org/10.5281/zenodo.7826791 (Cavaliere, 2023)

*Supplement: DOI*

*Author contributions.* Conceptualization, A.C, B.G.,F.C. and A.Z.; methodology, A.C, B.G.,F.C. and G.G.; software, A.C.; formal analysis, A.C., G.G. and B.G.; investigation, L.B., F.C., B.G. and A.Z.; data curation, A.C., F.C. and T.G.; writing—original draft preparation, A.C. and L.B.; writing—review and editing, A.C, L.B., G.G. and B.G.; visualization, B.P.A., M.S. and C.V.; project administration, A.Z., and C.V. All authors have read and agreed to the published version of the manuscript.

*Competing interests.* The authors declare that they have no conflict of interest.



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
