# Peer review of "Development of Low-Cost Air Quality Stations for Next Generation Monitoring Networks: Calibration and Validation of NO2 and O3 Sensors"

_EGUsphere, 2023_

## Author Comment (AC1)

We would like to thank the reviewer for the constructive comments. We have tried to address these comments in the attached response document, in the manuscript and in the code. Reviewer comments are reproduced in black, our responses are in blue.
* * *
General Comments:

This manuscript details the calibration process of low-cost metal oxide NO2 and O3 sensors. The authors evaluated the performances of several univariate, multivariate, linear, and non-linear calibration models. For these models the authors also analyzed the impact of individual predictors on model performance. The authors reccommended using multiple covariates in multiple regression models and to analyze the importance of the features used. Additionally, that machine learning models can greatly improve accuracy but have a harder time on data outside the calibration dataset.

Specific Comments:

At times the novelty of the approach seems to be overstated. Line 67 and 298 talk about the use of internal temperature as a calibration factor. Off the shelf sensors such as the Clarity Node S (measures NO2 with an electrochemical cell) use RH and internal temperature to adjust their NO2 readings). Additionally, in Line 221 you state that there is no statisical difference between using internal or external temperature. On line 298 you reference Figure 4 to explain why internal temperature was chosen but you do not show the same analysis for external temperature or for NO2.

Thanks for drawing attention to this aspect. As referenced in the study by Miech, Jason A., et al., (2021), the Clarity Node S adopts an electrochemical cell to measure NO2 levels. Notably, the NO2 measurement process is particularly intriguing, as the Alphasense NO2-A43F electrochemical cell comes equipped with a crucial ozone filter at its front end.
To further enhance precision, the device accounts for internal relative humidity and internal temperature while adjusting the NO2 measurements. Despite the comprehensive approach to NO2 measurement, the study does not explicitly specify the specific sensor used to measure environmental variables.

In our study, the correlation matrix and Bland-Altman plots revealed a robust positive correlation between the external and internal temperatures, with a relatively constant delta of 8°C (due to the significant heat radiation from the MOSs, causing the internal temperatures to be significantly higher than ambient conditions). The strong multicollinearity suggests that using both temperatures together would not be useful. Therefore, it would be appropriate to interchangeably use either of the temperatures.

As pointed out by the reviewer, upon examining Table 2 we observed a small yet minimal difference in using the internal temperature. However, we proceeded with the analysis using the internal temperature based on insights from the literature, such as the study by Schmitz, Seán, et al. (2021). This research emphasizes the importance of use internal temperature and relative humidity as essential factors because they more accurately represent normal MOS operating conditions.

In conclusion, we recognized the multicollinearity issue and opted to use only one of the temperatures in our analysis. Additionally, the use of external temperature as depicted in the figure below (fig. 1), does not lead to any definitive conclusions beyond what has already been discussed. Consequently, the highlighted paragraphs, as pointed out by the reviewer, have been revised to enhance clarity and improve its contribution to the overall analysis.

[Figure]

Fig. 1: Relationship between O3 raw data and extT in K-means cluster for AQ1 (a) and AQ2(b) stations. Regression lines were fitted to each cluster.

**References**

- Miech, J. A., Stanton, L., Gao, M., Micalizzi, P., Uebelherr, J., Herckes, P., & Fraser, M. P. (2021). Calibration of low-cost no2 sensors through environmental factor correction. Toxics, 9(11), 281. https://doi.org/10.3390/toxics9110281

- Schmitz, S., Towers, S., Villena, G., Caseiro, A., Wegener, R., Klemp, D., Langer, I., Meier, F., and von Schneidemesser, E. (2021). Unravelling a black box: an open-source methodology for the field calibration of small air quality sensors. Atmospheric Measurement Techniques, 14(11), 7221-7241. https://doi.org/10.5194/amt-14-7221-2021

The last paragraphs of introduction are then restructure on LL67-73 as:

These goals have been pursued by using ten among parametric, non-parametric univariate and multiple algorithms. Additionally, the investigation focused on delving deeper into the influence of internal temperature on LC sensors. To ensure comprehensive analysis, the covariate set for the multiple models was expanded to incorporate other essential factors such as humidity and gaseous interference compounds.

The paragraph of Discussion are then restructure on LL298-230 as:

Moreover, taking into account the observed multicollinearity issue between temperatures and the slightly higher mean accuracy, as well as the lower mean RMSE observed when using the internal one, the study drew upon insights from existing literature to identify the most suitable set of covariates (e.g., Schmitz, Seán et al., 2021; Miech, Jason A. et al., 2021). As a result, the inclusion of internal temperature as a significant factor was given priority, as it offers a more accurate representation of the operating conditions of the MOSs within the system. This approach was also adopted to tackle potential challenges in the board's analog-to-digital converter circuit.

Section 2.3: Please include more information on the sensor pre-deployment calibration with the HORIBA instruments. It is unclear whether this calibration was conducted indoors or outdoors or the spatial relationship between the AQ stations and the HORIBA instruments. If indoors please explain the lab environment where testing occurred.

As mentioned by the reviewer, it is unclear whether the calibration was conducted indoors or outdoors or the spatial relationship between the AQ stations and the HORIBA instruments. To address this, setup details are explained below.
The AQs are installed outdoors at the same height, securely mounted on a dedicated rack as shown in Figure 2a. Meanwhile, the HORIBA instruments are positioned indoors within the laboratory setting. To ensure an accurate representation of outdoor air conditions, two sampling probes, each approximately two meters in length and equipped with rain covers, are employed. These probes collect the outside air and channel it directly to the reference instruments (as depicted in Figure 2b). This setup enables us to sample and compare the air quality data collected by the AQs with the measurements obtained by the HORIBA instruments. For completeness we included Figure 2 below.

[Figure]

Fig. 2: AQs on the left (a); Horiba setup on the right (b).

The paragraph of Section 2.3 are then restructure on LL298-230 as:

As detailed in Table S1 of the Supplementary material, Pre–deployment calibration of AQ1 and AQ2 stations against HORIBA analyzers was performed at CNR-IBE headquarters in Florence, Italy (43°47'52" N, 11°11' E, Figure 1). The AQ stations were mounted on a dedicated outdoor rack, while the HORIBA instruments were placed indoors in a laboratory setting. For outdoor air pollution sampling, approximately two–meter–long sampling probes were employed to collect outside air and channel it directly to each of the reference instruments.

Line 196: Do you have any explanation for why more data was withdrawn from AQ2 compared to AQ1?

We appreciate the reviewer for bringing attention to this difference between the AQs. We identified that AQ1 had a 2% withdrawal due to the MOS NO2 sensor, while AQ2 had a 12% withdrawal, primarily from RH (7%) than from MOS NO2. It's worth mentioning that the RH sensor in AQ2 reached saturation (data > 99%) more frequently than in AQ1, impacting the overall data.

Figure 8a: Should this legend read "AQ1 O3 MLR" rather than NO2?

We thank the reviewer for pointing out this typo, we have fixed it in the manuscript.

While Figure S10 summarizes the NO2 and O3 concentrations across the validation period and Table 6 for the field validation it would also be useful to include a table detailing the historical environmental conditions of both the field validation and calibration period, such as RH and temperature. This could help support the points made in line 340, as when the environmental conditions differ between pre-deployment calibration and the deployment/validation period the MRF model may suffer.

Table S1 now includes environmental parameters, namely temperature and relative humidity. Additionally, for each process, we have specified the reference intervals, Horiba for the pre-deployment calibration and ARPAT for the field validation. We also corrected an error in the date interval of pre-deployment.

Line 331: Please re-word this sentence as the point is unclear

Thanks for the suggestion.

Line 331  it has been corrected as in LL331-333:

Ultimately, this resulted in robust LC performances outside the training conditions and the ability for easy adjustments to cope with changes in sensor performance over time.

Line 339: You mention global impacts of this analysis but provide no other information of how this work extends to beyond Italy.

Thanks for the suggestion. In the ongoing activity, the AIRQino LC stations are planned to be deployed outside Italy and also in extreme environmental conditions (Carotenuto et al., 2020). This will allow NO2 and O3 sensors to be tested under different meteorological and air pollution conditions. The conclusions now include some details of ongoing projects and perspective.

**Reference**

- Carotenuto, Federico et al.,  2020 IOP Conf. Ser.: Earth Environ. Sci. 489 012022. https://doi.org/10.1088/1755-1315/489/1/012022

The conclusions now include some details of ongoing project and perspective, as follows (LL349-end):

A limitation of the present work is that the LC stations have been calibrated during a period not particularly long (70 days) and a typically summer one, thus when pollution levels are generally meaningful for O3, but they are not for NO2 concentrations. Indeed, conducting a pre–deployment calibration during a winter period, when NO2 concentrations are typically higher, would be a valuable addition to the study. This step would provide a more comprehensive understanding of the AQs validation performance under varying pollution conditions and help address the limitation of the current calibration period biased towards summer data. Moreover, conducting a similar validation outside of Italy, in regions with differing pollution and meteorological conditions would be of great interest. For this purpose, in the ongoing activity, the AIRQino LC stations are planned to be deployed outside Italy, such as in Nice and Aix–en–Provence (France), Barcelona (Spain), Budapest (Hungary), Tirana (Albania), and Niamey (Niger).

Furthermore, in the future, a new sensor for monitoring NO could hopefully be integrated into the LC stations and validated. As such, the combined monitoring of NO, NO2 and O3 concentrations and their daily and seasonal variability would allow a comprehensive pattern of the oxidant capacity of the atmosphere, particularly effective in southern Mediterranean countries such as Italy (Pancholi et al., 2018). In addition, once the AQ VOC sensor is validated, it will enable the monitoring of all O3 precursors (VOC and NOx). This comprehensive monitoring, combined with the application of SHapley Additive exPlanations (SHAP) method, will lead to a full characterization of photochemical pollution in various areas of interest, including urban, sub-urban, or rural regions. Moreover, portability of LC sensors makes them ideal devices for filling knowledge gaps in regions that are difficult to access such as the open sea. Mounted on buoys or ships, for example, LC sensors could collect the high O3 levels that typically occur over these areas in summer due to high solar activity and rather low mixing height combined with a lack of O3–consuming NO emissions.

**References**

- Pancholi, P., Kumar, A., Bikundia, D. S., & Chourasiya, S. (2018). An observation of seasonal and diurnal behavior of O3–NOx relationships and local/regional oxidant (OX= O3+ NO2) levels at a semi-arid urban site of western India. Sustainable Environment Research, 28(2), 79-89. https://doi.org/10.1016/j.serj.2017.11.001

- Li, W., Wang, Y., Liu, X., Soleimanian, E., Griggs, T., Flynn, J., & Walter, P. (2023). Understanding offshore high-ozone events during TRACER-AQ 2021 in Houston: Insights from WRF-CAMx photochemical modeling. EGUsphere, 2023, 1-21. https://doi.org/10.5194/egusphere-2023-1117.

---

## Author Comment (AC2)

We would like to thank the reviewer for the constructive comments. We have tried to address these comments in the attached response document, in the manuscript and in the code. Reviewer comments are reproduced in black, our responses are in blue.
* * *
Overall, this effectively communicates the full pipeline of data collection, calibration, and validation of NO2 and O3 low-cost sensors. The authors have shown a clear commitment to transparent science and carefully applied data science principles while staying relevant to the domain of atmospheric science. Importantly, rather than just fitting a plethora of models, the overall interpretability of features is investigated – including how the relevance of features varies across different meteorological and pollutant loading regimes over the course of a relatively long timeline. However, this paper suffers from several structural weaknesses. It overstates the novelty of applying SHAP and generally should cite more recent literature throughout. Furthermore, the novelty of exploring feature relevance is somewhat lost in the acronym dense model performance metrics – most of which are already well characterized in literature. After restructuring the paper to more precisely state and describe its novel findings, it will be a useful reference for the community.

**Introduction**:

Overall, the introduction should be better organized. It would be helpful to first motivate applications of low-cost sensors with a short (1-2 sentences) summary of the relevance of fine-scale spatial-temporal NO2 and O3 patterns before diving into their design. Consider citing work like: doi.org/10.1016/j.envint.2018.04.002, doi.org/10.1136/bmj.n534. Don't use phrases like "in the last few years" or "nowadays" (line 23) – Be specific on when low-cost sensors emerged and when deployments scaled.

Low-cost sensors for air quality monitoring emerged in the early 2010s, with deployment scaling up in the mid to late 2010s. These sensors offer a cost-effective solution for monitoring air pollutants like NO2 and O3, enabling widespread deployment and data collection. Their ability to reveal fine-scale spatial-temporal patterns of NO2 and O3 is valuable for identifying pollution hotspots, evaluating air quality control measures, and supporting targeted interventions to mitigate health risks from air pollution. As pointed out by the reviewer, this information, including the mentioned works, was explicitly stated at the beginning of the introduction.

This information has been explicitly mentioned in the text, specifically on LL20-26:

Low-cost (LC) air quality sensors are gaining more and more interest as they can provide near real-time observations with high spatial and temporal resolution. Their observations can be integrated into the current official regulatory networks, usually monitoring air quality at lower space and time resolution, thus providing useful information to support policymakers and stakeholders in understanding air pollution dynamics (Brilli et al., 2021; Morawska et al., 2018). Dramatic advances in LC sensor technology have been made since their very first applications for monitoring CO, NO2 and NOx (De Vito et al., 2009), O3 (Williams et al.,

2013), and particulate matter (Holstius et al., 2014). Among gaseous species, NO2 and O3 are the most commonly investigated since both short- and long-term exposure to these pollutants are associated with higher risk to human health (Linares et al., 2018; Nuvolone et al., 2018; Meng et al., 2021; World Health Organization, 2021).

**References**

- De Vito, S., Piga, M., Martinotto, L., & Di Francia, G. (2009). CO, NO2 and NOx urban pollution monitoring with on-field calibrated electronic nose by automatic bayesian regularization. Sensors and Actuators B: Chemical, 143(1), 182-191. https://doi.org/10.1016/j.snb.2009.08.041

- Holstius, D.M., Pillarisetti, A., Smith, K.R., & Seto, E.J.A.M.T. (2014). Field calibrations of a low-cost aerosol sensor at a regulatory monitoring site in California. Atmospheric Measurement Techniques, 7(4), 1121-1131. http://doi.org/10.5194/amt-7-1121-2014

- Linares, C., Falcón, I., Ortiz, C., & Díaz, J. (2018). An approach estimating the short-term effect of NO2 on daily mortality in Spanish cities. Environment International, 116, 18-28. https://doi.org/10.1016/j.envint.2018.04.002

- Meng, X., Liu, C., Chen, R., Sera, F., Vicedo-Cabrera, A.M., Milojevic, A., et al. (2021). Short term associations of ambient nitrogen dioxide with daily total, cardiovascular, and respiratory mortality: multi location analysis in 398 cities. bmj, 372. https://doi.org/10.1136/bmj.n534

- Nuvolone, D., Petri, D., & Voller, F. (2018). The effects of ozone on human health. Environmental Science and Pollution Research, 25, 8074-8088. https://doi.org/10.1007/s11356-017-9239-3

- Williams, D.E., Henshaw, G.S., Bart, M., Laing, G., Wagner, J., Naisbitt, S., & Salmond, J.A. (2013). Validation of low-cost ozone measurement instruments suitable for use in an air-quality monitoring network. Measurement Science and Technology, 24(6), 065803. https://doi.org/10.1088/0957-0233/24/6/065803

The phrasing in Line 45 is confusing. I suggest removing the claim that there are "no established protocols" and instead merely stating there are two common strategies. There are some existing guidelines from relevant government agencies (cfpub.epa.gov/si/si_public_file_download.cfm?p_download_id=517654, as well as publications.jrc.ec.europa.eu/repository/handle/JRC83791)

As the reviewer suggests, instead of mentioning "no established protocols," we have provided clarification by stating that there are two common strategies for deploying low-cost sensors. Moreover, it is important to acknowledge the existing guidelines from relevant government agencies and scientific publications, which offer valuable insights into sensor deployment and data interpretation.

Line 45 it has been corrected as in LL44-45:
Two main approaches to calibrating LC sensors exist (Spinelle et al., 2013): Pre–deployment and field calibration.

**Reference**

- Spinelle, L., Aleixandre, M., Gerboles, M. (2013). Protocol of evaluation and calibration of low-cost gas sensors for the monitoring of air pollution. Luxembourg (Luxembourg): Publications Office of the European Union. https://doi.org/10.2788/9916.

Please restructure the last two paragraphs of the introduction as to not exaggerate claims of novelty. This would not be the first study to use SHAP for environmental low-cost sensor evaluation as the authors claim at the end of the introduction section: doi.org/10.1016/j.atmosenv.2023.119692, doi.org/10.3390/s20195497, & doi.org/10.1109/SENSORS52175.2022.9967180. Furthermore, although the 3 gaps identified by the authors in the literature are still relevant areas of investigation, please better contextualize them – for example (iii) has been an active area of investigation with many recent publications as referenced earlier regarding SHAP and low-cost sensors.

As mentioned by the reviewer, this study is not the first to work with SHAP, at least three other works have utilized SHAP and reported its application.

The first work cited (doi.org/10.1016/j.atmosenv.2023.119692) is particularly interesting; however, it is worth noting that it was published after the publication of this preprint. While the work presents the summary plot of SHAP values for electrochemical NO2 and PMs, it exclusively focuses on random forest models, omitting the exploration of the intriguing lightGBM algorithm. Additionally, the paper does not mention the type of explainer used (we can hypothesize a kernel one), which is crucial given its potential impact on computation time. The second work mentioned (doi.org/10.3390/s20195497) involves low-cost and low-resolution heat thermal sensors, making it indirectly related to pollution monitoring. Nevertheless, the paper's classification perspective, contrasting our regression approach, is fascinating. It applies SHAP to a features vector extracted from heat sensor images, employing the CatBoost classifier.

The third paper (doi.org/10.1109/SENSORS52175.2022.9967180) tackles this challenge by adopting diverse methodologies of explainable artificial intelligence (XAI) for gas sensors. Its exploration of global interpretability perspectives for SHAP Analysis in both Pulse and Sinusoidal Mode is particularly intriguing. Moreover, the paper refers to other XAI methods, including Local Interpretable Model-agnostic Explanation (LIME), but it does not address the issue of cross-sensitivity with environmental parameters.

By elaborating on these mentioned works, we have contextualized our study while highlighting the specific areas where our research contributed novel insights.

The last paragraphs are then restructure on LL67-75:

These goals have been pursued by using ten among parametric, non-parametric univariate and multiple algorithms. Additionally, the investigation focused on delving deeper into the influence of internal temperature on LC sensors. To ensure comprehensive analysis, the covariate set for the multiple models was expanded to incorporate other essential factors such as humidity and gaseous interference compounds. Furthermore, the study utilized

model-agnostic techniques, including SHapley Additive exPlanations (SHAP) introduced by Lundberg and Lee (2017), to assess the model's generalization ability in a field environment. While SHAP has been employed in previous pollution-related studies (e.g., Wang, An et al., 2023; Chakraborty, Sanghamitra et al., 2022; Vega García and Aznarte, 2020), our research provides an original contribution by applying SHAP specifically to MOS sensors. This application aims to provide both local and global interpretations, resulting in a deeper understanding of the sensor's behavior on individual data points and gaining insights into its overall performance.

**References**

- Wang, A., Machida, Y., deSouza, P., Mora, S., Duhl, T., Hudda, N., ... & Ratti, C. (2023). Leveraging machine learning algorithms to advance low-cost air sensor calibration in stationary and mobile settings. *Atmospheric Environment*, *301*, 119692. https://doi.org/10.1016/j.atmosenv.2023.119692

- Chakraborty, S., Mittermaier, S., Carbonelli, C., & Servadei, L. (2022, October). Explainable AI for Gas Sensors. In *2022 IEEE Sensors* (pp. 1-4). IEEE. http://doi.org/(doi.org/10.1109/SENSORS52175.2022.9967180

**Materials and Methods:**

Clustering analysis does not obviously follow from a correlation analysis – especially if it is expected that many environmental variables will be collinear. Additionally, this paper emphasizes the importance of trying many supervised methods but offers no justification for the unsupervised method – this is especially tricky since K-means is probably not the best method for identifying robust clusters given the expected collinearity.

As the reviewer pointed out, clustering analysis may not be an obvious consequence of a correlation analysis. Moreover, interpreting the K-means findings necessitates domain knowledge and careful consideration of collinearity between variables, especially when employing multidimensional clustering algorithms. In our work, we utilize a 1D k-means clustering as a standalone tool, enabling us to gain valuable insights into the distribution of data in particular O3 concentrations.

The exploration of the potential generalizability of the relationship between sensor data and temperature, which is also evident in the correlation matrix, was initiated by conducting a preliminary qualitative analysis. Within this analysis, we employed clustering to group the Horiba O3 concentrations and utilized these clusters to create a color mapping within a scatter plot, effectively visualizing the relationship between sensor raw data and internal temperature. Subsequently, we applied linear regression to determine the direction of the relationship between sensor raw data and temperature variables within each group. Notably, the inclination of the regression lines remained relatively constant across the clusters, indicating the possibility of developing a generalized model by leveraging this consistent relationship.

It is worth noting that k-means clustering on 1D data can be solved in polynomial time, making it computationally efficient. However, this method shows limitations as any other one, and the reviewer exposed some of them in a later section where we tried to respond.

The use of the unsupervised method is then justify better on LL106 (ex 104)-116:

Prior to implementing the calibration techniques, an exploratory data analysis (EDA) was conducted using the correlation matrix to identify important insights. The study further explored the potential generalizability of the relationship between MOS O3 sensors and temperature, as highlighted in Spinelle et al. (2016), leveraging observations from the correlation matrix. To gain deeper insights, a preliminary qualitative analysis involved employing one-dimensional K-means clustering (MacQueen,J, 1965) to group the Horiba concentrations. These clusters were visually represented using color mapping within a scatter plot, effectively illustrating the connection between sensor raw data and internal temperature. Drawing on the visual insights from the clustering analysis, linear regression was implemented to investigate the direction of the relationship between sensor raw data and temperature variables within each cluster. To determine the optimal number of clusters (k) for the analysis, the elbow method of the distortion metrics was employed (Bengfort et al., 2018). This step yielded preliminary insights into the relation of sensor data and temperature variables within distinct clusters, further confirming the decision to pursue a more comprehensive examination using multivariate models for pre-deployment calibration.

This section is very acronym and initialism heavy please consider writing out at least some of the less utilized terms. It may also enhance readability to move many of the details describing the exact model instantiations and hyperparameters (2.3.1 & 2.3.2) employed to a table in the appendix or SI – especially since many of the models are "off-the-shelf" from scikit-learn and not developed by the authors.

Thanks for the suggestions. We have addressed them by spelling out less used acronyms and initialisms, including those for environmental parameters and some of the multiparametric algorithms. Furthermore, to enhance the comprehensiveness of the document, we have now incorporated Table S2, which contains detailed information about model instantiations and hyperparameters.

SI Table 1 would benefit from also including some historical data about pollution concentrations from NO2 and O3.

In Table S1 we have now included environmental parameters, namely temperature and relative humidity. Additionally, for each process, we have specified the reference intervals, Horiba for the pre-deployment calibration and ARPAT for the field validation.

**Results**:

I recommend changing the joint plots in Figure 3 from hex-binned heatmaps to the much more intuitive scatterplots.

Figure 3 was updated from a hex-binned heatmap to a more intuitive scatterplot.

The purpose of the k-means analysis is still unclear here. These 6 clusters maybe the most robust set k-means could identify, but that does not mean they are a meaningful or interpretable clustering. From Figure 4, there does not seem to be an obvious regime change or large Euclidean distance between clusters. I would recommend removing these results or considering a density-based clustering approach.

The optimal number of clusters (6) was determined using k-means, with the Elbow method serving as the guiding metric based on the distortion score. The resulting clusters exhibited a distance of approximately 30 µg/m3 between centroids. However, due to the time sampling of data collection in a relatively stable summer condition, we did not anticipate significant Euclidean distances between the k-means clusters. Furthermore, the data's distribution, along with the one-dimensional approach, made density-based clustering methods unsuitable for our specific purpose, resulting in a single point cloud.

However, if the reviewer finds it appropriate, the alternative to completely removing figure 4 could be to adopt a classification rather than a clustering approach. One potential replacement for the k-means analysis could be the Jenks optimization method, also known as the Fisher-Jenks algorithm. This method can be used to identify "natural breaks" within the data, allowing to calculate Horiba class boundaries more effectively. The Jenks optimization method is also known as the goodness of variance fit (GVF) and It is used to minimize the squared deviations of the class means (Jenks, G. F. 1967). Geographic Information Systems (GIS) and related geospatial software products often employ the "natural breaks" classification for interpreting pollution and emission variability (e.g., De Smith, Michael John, et al., 2007; Carabetta, João Luiz Martins 2019; Xia, Qi, et al., 2022). By incorporating the Jenks optimization method and binning the Horiba data concentrations, we aim to visualize the results using a categorical palette, as suggested by the reviewer. This approach will lead to an improved Figure 4, which will present the outcomes in a more comprehensive and informative manner. The considerations made for the part on the regressions remain valid. In this case, the code to obtain the class boundaries using the Jenks optimization method will be added to the GitHub repository. Additionally, we will replace the k-means section in the Material and Methods section to reflect the use of the Jenks optimization method.

[Figure]

Fig. 4: Relationship between O3 raw data and intT in HORIBA O3 natural breaks classes for AQ1 (a) and AQ2 (b) stations. Regression lines were fitted to each cluster. The coefficients for each regression line and the upper bound of each class are reported in the legend.


Similarly, the regression lines in Figure 4 are not obviously interpretable, they seem noisy and less robust than simply relying on correlation matrix. I would suggest promoting the bottom triangles of the two Pearon's r matrices in SI Fig S2 and removing Figure 4. If Figure 4 stays in the manuscript, it should also use a different colormap, the yellow does not appear clear on my computer screen. I would recommend a categorical colormap such as Hawaii from https://www.fabiocrameri.ch/colourmaps/.

As mentioned in the reviewed version of materials and methods, we conducted regression analyses for each class of Horiba concentration to represent the relationship between count and temperature within each group. As pointed out by the reviewer, the regression lines are not obviously interpretable. However, the qualitative analysis showed that the inclination of the regression lines remained nearly constant across the clusters, indicating a consistent trend in the relationship between count and temperature within each class of Horiba concentration. This suggests that the relationship between count and temperature within each group is relatively stable.

The presence of a noticeable deviation in AQ2, particularly in correspondence with the 0-39 µg/m³ interval of O3 concentration, suggests that there is a distinct relationship between count and temperature in this specific range of concentrations. At this concentration it's possible that the sensor may be operating at its lower limit, which can lead to nonlinear responses.

Figures 5 & 6 are very useful for telling the story of your paper. Consider enhancing them by increasing the height of the y-axis or adding jitter to the points to avoid the overlap as it adds to visual clutter. Furthermore, please change the colormap as suggested for Figure 4.

To address points overlap, we have increased the y-axis range in Figure 5 and 6. Additionally, in response to the reviewer's suggestion, we have migrated the figures (Fig. 4, Fig. 5, Fig. 6 and Fig. 7) to two main palettes, guided by the provided resource:

1. Cubehelix: This sequential palette was selected due to its smooth transition from dark to light, making it suitable for representing continuous or ordered data.
2. Seaborn's Colorblind: We chose this categorical palette as it is designed specifically for representing discrete or categorical data.

Both are designed to be easily distinguishable for readers with colour vision deficiencies.

While Taylor plots like those in Figure 8 can be useful, it seems a table would more succinctly get the point across. I'd recommend moving it to the supplement.

The Taylor diagrams were replaced with a summary table, now presented as Table 6, while the original Taylor diagrams were moved to the supplementary materials as Figure S10.

**Discussion & Conclusions**

The first two paragraphs of the Discussions section can be combined and made more concise. They do not communicate the novelty of the study and reflect an overall structural problem with this manuscript – too much emphasis is placed on the individual "off-the-shelf" models rather than the much more interesting implications of feature relevance at differing concentration regimes or the role of model complexity in spatial-temporal transferability.

Thanks for the suggestion. In response, we made efforts to condense the initial two paragraphs while incorporating a concise comparison of the literature findings. This allowed us to focus on exploring the implications of feature relevance across various concentration regimes and the role of model complexity in spatial-temporal transferability.

The discussions are then restructure starting from L279 as follow:

[revised manuscript text omitted]

The discussion point on seasonal transferability is quite interesting and I recommend expanding on it. The comparison of DA, PFI, and SHAP is out of place and would be much better in the methods section with references to literature.

Thanks for incorporating the suggestion. We have now moved the comparison of DA, PFI, and SHAP to section 3.4, allowing readers to grasp the differences in these approaches right away. Furthermore, in response, we have expanded the discussion on seasonal transferability in the last paragraphs.

The final paragraphs of the discussion section now encompass a detailed expansion on the topic of seasonal transferability, as follows:

The seasonal analyses presented in Table 7 provided an overview of these seasonal changes in the stability and biases of AQ1 and AQ2 O3 and NO2 sensors for the application of the MLR calibration model after deployment. O3 pre-deployment calibration showed good performance in all seasons and for both stations the lowest nRMSE value was registered in summer 2018 and in summer 2019 and the highest value of nRMSE was recorded in winter 2018-2019. The decline in performance during the winter period was minimal, despite the fact that the pre-deployment calibration was mainly performed in the summer.
Furthermore, the comparison of the summer period of 2018 and 2019 showed a decrease of 2% in nRMSE for both AQs and pollutants (O3 and NO2). The decrease in nRMSE for O3 was accompanied by an increase in the magnitude of MAE and MBE, pointing towards a possible linear drift in O3 sensor readings after a year of use. Conversely, for pre-deployment calibration of NO2, a decrease in MBE was observed. The decrease in MBE

for NO2 and the prominent role of O3 raw readings and its negative impact on prediction, as identified through feature importance analysis of the pre-deployment MLR model, further reinforced the idea of a linear drift in O3 sensor readings. Similarly, the pattern of lowest and highest nRMSE values for O3 validation remained consistent also for the MRF model, being the lowest in the summer of 2018 and 2019 and the highest in the winter of 2018-2019 (Table S12). Notably, AQ1 outperforms AQ2 in both models; however, as mentioned earlier, the differences in nRMSE values between the MLR and MRF models were quite significant.

The conclusions should include some detail about implications for work outside of Italy, in differing pollution and meteorological regimes.

Thanks for this valuable suggestion. AIRQino LC stations are installed outside for ongoing projects. Furthermore the deployment of AIRQino LC stations has expanded not only beyond Italy but also to extreme environments, such as the Arctic region, as demonstrated by Carotenuto et al., (2020). This extension has opened new avenues for ongoing research, enabling the comprehensive study of diverse pollution and meteorological patterns across various regions. In continuation of these efforts, three AQs have recently been installed in Niamey, Niger. This new deployment holds significant importance, as Niamey's climate conditions can be extreme, characterized by high temperatures and unique challenges.

**Reference**

- Carotenuto, Federico et al.,  2020 IOP Conf. Ser.: Earth Environ. Sci. 489 012022. https://doi.org/10.1088/1755-1315/489/1/012022

The conclusions now include some details of ongoing project and perspective, as follows (LL349-end):

A limitation of the present work is that the LC stations have been calibrated during a period not particularly long (70 days) and a typically summer one, thus when pollution levels are generally meaningful for O3, but they are not for NO2 concentrations. Indeed, conducting a pre–deployment calibration during a winter period, when NO2 concentrations are typically higher, would be a valuable addition to the study. This step would provide a more comprehensive understanding of the AQs validation performance under varying pollution conditions and help address the limitation of the current calibration period biased towards summer data. Moreover, conducting a similar validation outside of Italy, in regions with differing pollution and meteorological conditions would be of great interest. For this purpose, in the ongoing activity, the AIRQino LC stations are planned to be deployed outside Italy, such as in Nice and Aix–en–Provence (France), Barcelona (Spain), Budapest (Hungary), Tirana (Albania), and Niamey (Niger).
Furthermore, in the future, a new sensor for monitoring NO could hopefully be integrated into the LC stations and validated. As such, the combined monitoring of NO, NO2 and O3 concentrations and their daily and seasonal variability would allow a comprehensive pattern of the oxidant capacity of the atmosphere, particularly effective in southern Mediterranean countries such as Italy (Pancholi et al., 2018). In addition, once the AQ VOC sensor is validated, it will enable the monitoring of all O3 precursors (VOC and NOx). This comprehensive monitoring, combined with the application of SHapley Additive exPlanations (SHAP) method, will lead to a full characterization of photochemical pollution in various

areas of interest, including urban, sub-urban, or rural regions. Moreover, portability of LC sensors makes them ideal devices for filling knowledge gaps in regions that are difficult to access such as the open sea. Mounted on buoys or ships, for example, LC sensors could collect the high O3 levels that typically occur over these areas in summer due to high solar activity and rather low mixing height combined with a lack of O3–consuming NO emissions.

Consider promoting SI Figure S10 to the main text, it is useful for understanding the discussion points as well as contextualizing the range of pollutant concentrations of this study.

As suggested by the reviewer, we have now incorporated Figure S10 into the main text and renumbered it as Figure 8.